# Effect of Impregnated Phenolic Resins on the Cellulose Membrane for Polymeric Insulator

**DOI:** 10.3390/membranes12020106

**Published:** 2022-01-18

**Authors:** Sharifah Nurul Ain Syed Hashim, Sarani Zakaria, Chin Hua Chia, Zalita Zainuddin, Thomas Rosenau, Sharifah Nabihah Syed Jaafar

**Affiliations:** 1Department of Applied Physics, Faculty of Science and Technology, Universiti Kebangsaan Malaysia, Bangi 43600, Selangor, Malaysia; nur_ain9087@yahoo.com (S.N.A.S.H.); szakaria@ukm.edu.my (S.Z.); chia@ukm.edu.my (C.H.C.); zazai@ukm.edu.my (Z.Z.); 2Division of Chemistry of Renewable Resources, Department of Chemistry, University of Natural Resources and Life Sciences Vienna, Muthgasse 18, A-1190 Vienna, Austria; thomas.rosenau@boku.ac.at; 3Johan Gadolin Process Chemistry Centre, Åbo Akademi University, Porthansgatan 3, FI-20500 Åbo/Turku, Finland

**Keywords:** CP-MAS NMR, crosslink, empty fruit brunch, resistivity, surface roughness

## Abstract

In this study, a cellulose membrane (CM) was chemically treated with phenolic (PF) resin to improve its performance as a polymeric insulator. The CM was prepared from kenaf pulp, and the PF was synthesized from oil palm empty fruit (EFB) fibre. Four different concentrations of synthesized PF resin (5, 10, 15, and 20 wt.%) were impregnated under wet or dry conditions. Thermal analysis of the phenolic cellulose membrane (PCM) showed that the samples had good chemical interaction and compatibility. The PF uptake in the wet phenolic cellulose membrane (PCMW) was higher than in the dry phenolic cellulose membrane (PCMD). During the PF uptake, the CM underwent solvent exchange and absorption in wet and dry membranes, respectively. This difference also affected the crosslinking of PCM samples via the formation of methylene bridges. Due to the PF treatment, the PCM showed lower water absorption than CM. The PF concentrations also affect the surface roughness and electrical properties of PCM samples. These findings prove that PCM can be used as a renewable and green polymer electrical insulator.

## 1. Introduction

Polymers, such as ethylene propylene rubber (EPR) [1], ethylene propylene diene methylene (EPDM) [2], silicon rubber [3], paper [4], and thermoset resins, such as phenolic resin (PF) [5], are commonly used for electrical insulation, in addition to glass and ceramic. The polymer electrical insulator is widely used in electronic packaging [6], thermal management [7], high-voltage systems [8], coating [9], distribution lines [10], etc. Polymer insulators, also known as composite insulators, are suitable for use in high-voltage (HV) products because of their ability to increase the charge capacity. The advantages of this type of insulator include its light weight, low cost, superior contamination performance, high resistivity, and better flexibility, thus making it a suitable replacement for other materials. However, the performance of polymer insulators will decline and fall over time, due to factors that include exposure to excessive moisture and heat, natural aging, chemical deterioration, mechanical damage, etc.

Cellulose is seen as a potential candidate in thermal and electrical insulation applications, due to its excellent properties that include an easy manufacturing process and high thermal stability [11]. Huang et al. (2017) found that the introduction of 10 wt.% of nano-fibrillated cellulose (NFC) not only enhanced the breakdown performance, but also improved its thermal and mechanical properties [12]. The permittivity of electrical grade pulp approached 5.3, and tan δ lies between 0.01 and 0.02 at 50 Hz. However, to achieve this, the preparation of the raw material plays an important role because they contain different chemical compositions, which, in turn, lead to different electrical insulating properties [13]. Brzyski et al. (2019) obtained cellulose with thermal conductivity ranging from 0.041 to 0.047 W/(m·K), and with a density ranging from 63 to 43 kg/m^3^ [14]. This showed that the thermal conductivity of the cellulose-based insulator can be modified by altering the density of the cellulose fibre. Furthermore, the addition of nanoparticles, such as silicon dioxide, titanium oxide, and montmorillonite, can mitigate current breakdown.

Even though cellulose-based materials have been extensively used as insulators, they are moisture absorbing and show low electronic polarisation. Consequently, insulation properties cannot be maintained permanently, and short circuits might occur. Alternatively, surface modification is an option to overcome the problem, and may also be able to enhance the dielectric properties; for example, the incorporation of phenolic (PF) resin into cellulose would reduce the cellulose hydrophilicity. PF resin is a thermoset resin that is resistant to chemicals and moisture over a long period [15]. Its chemical structure allows the PF to have high polarizability and good thermal properties that can withstand temperatures up to 380 °C [16,17]. Moreover, PF resin is a non-conductive material due to its low charge carrier concentration [18,19].

In this study, we focused on improving the electrical properties of the cellulose membrane (CM) by impregnation with PF resin. The impregnated CM, named phenolic cellulose membrane (PCM), was prepared under wet or dry conditions. More extensive knowledge regarding the utility of PCM polymer insulators was achieved from understanding the influence of the impregnation mechanism and the PF resin uptake on their physical, chemical, thermal and surface properties. Our findings are important steps towards developing a green, renewable, and sustainable polymer insulator for electronic and thermal applications. 

## 2. Experimental Procedure

### 2.1. Materials

Commercial resole PF (CK-1634) was obtained from Malayan Adhesives & Chemicals. Oil palm empty fruit bunch (EFB) fibre was obtained from Szetech Engineering Sdn. Bhd. (Shah Alam, Selangor, Malaysia), kenaf core pulp was obtained from Malaysian Agricultural Research and Development Institute (MARDI) (Serdang, Selangor, Malaysia); meanwhile phenol (95%), formalin (37%), sulfuric acid (H_2_SO_4_) (95–98%), methanol (99%) and sodium hydroxide (NaOH) (99%) were purchased from R&M Chemicals (Ever Gainful Enterprise Sdn Bhd, Selangor, Malaysia). 

### 2.2. Preparation of Cellulose Membrane (CM)

NaOH/urea solvent was prepared by using NaOH, urea and distilled water with a weight percentage ratio of 7:12:81 and stored in a freezer (−13 °C). Kenaf core cellulose (4 wt.%) was dissolved in the cold NaOH/urea solvent and stirred at 2300 rpm with a mechanical stirrer. The cellulose solution was then centrifuged for 5 min at 10,000 rpm at 5 °C. The cellulose solution was poured onto the glass plate before being coagulated in 5% of H_2_SO_4_. After 15 min of coagulation, the formed CM was washed three times with distilled water and then freeze-dried. The freeze-dried CM was denoted as dried CM, and the non-freeze-dried CM was denoted as wet CM.

### 2.3. Preparation of PF Resin

Liquefied empty fruit bunch (LEFB) was prepared by mixing a 1:3 ratio of EFB fibres to phenol in the presence of 15 g of H_2_SO_4_ as catalyst and heated for 90 min at 150 °C. Methanol was then added, followed by filtration and evaporation process to obtain pure LEFB. Synthesized resol PF was prepared by mixing LEFB with formaldehyde at 60 °C for the first 1 h and 80 °C for the next 1 h. Then, 40% NaOH was slowly added into the mixture each hour, to maintain the pH at 9. The percentage yield of LEFB was calculated for dry conditions.

### 2.4. Fabrication of PCM Materials

The synthesized PF was diluted into four different concentrations (5, 10, 15 and 20 wt.%) with methanol. Then, the dried and wet CMs (1 cm · 1 cm) were impregnated with the PF at room temperature. After 1 h of impregnation, the CMs were placed between two pieces of Teflon sheet and loaded with 1 kg wooden block to avoid shrinkage and deformation. The samples were then cured in an oven at 115 °C for 24 h. The dried and wet cured samples were denoted as PCMD and PCMW, respectively.

### 2.5. Chemical Characterizations

Dried residue and EFB fibres were analysed for their lignin and holocellulose compositions by using TAPPI Standard Method T222. The chemical functionalities of PF resin, CM and PCM materials were analysed using Fourier transform infrared (FTIR) and cross polarization magic-angle-spinning nuclear magnetic resonance (CP-MAS NMR) spectroscopy, respectively. The FTIR (Bruker ATR-FTIR Alpha System, Bruker, Billerica, MA, USA) was performed with 60 scans in 4 cm^−1^ resolutions in a range of 4000–650 cm^−1^ transmittance mode. The FTIR spectra were replotted using SigmaPlot 12.0 (Systat Software Inc. (SSI), San Jose, California, Version 12). The CP-MAS NMR was performed using a Bruker AVANCE III HD 400 spectrometer (Bruker, Bruker, Billerica, MA, USA) (resonance frequency of 400.13 MHz and 100.61 MHz for ^1^H and ^13^C, respectively), equipped with 4 mm dual-broadband CP-MAS probe. ^13^C spectra were acquired by using the TOSS (total sideband suppression) sequence at ambient temperature with a spinning rate of 5 kHz, a cross-polarization (CP) contact time of 4 ms, a recycle delay of 2 s, SPINAL-64 ^1^H decoupling and an acquisition time of 49 ms, whereas the spectral width was set to 250 ppm. Chemical shifts were referenced externally against the carbonyl signal of glycine with δ = 176.03 ppm.

### 2.6. Thermal Characterizations

Thermal analysis of PF resin was performed using thermogravimetric analysis (TGA) (Hitachi STA7000 Series, Hitachi High Technologies, Tokyo, Japan) and differential scanning calorimetry (DSC) (Perkin Elmer, Waltham, MA, USA). The tests were performed by heating each sample in a crucible up to 800 °C with a heating rate of 10 °C min^−1^ under nitrogen atmosphere. The CM and PCM materials’ thermal analyses were performed under the same atmosphere and heating rate, but used different brands of TGA and DSC, which were Shimadzu TGA50 (Shimadzu, Kyoto, Japan) and Mettler Toledo 822e (Columbus, OH, USA). The samples were heated from room temperature up to 600 °C for TGA and 300 °C for DSC.

### 2.7. Water Absorption, Surface, and Electrical Characterizations of PCM Materials

The water absorption test of CM and PCM materials were carried out according to ASTM D570–98. The sample was dried at 105 °C, then weighed (dry weight) and fully immersed in water for 24 h. The water on the sample’s surface was wiped and reweighed (wet weight). The water absorption was calculated using the following Equation (1):(1)$$\mathrm{Water}\;\mathrm{absorption}=\frac{\mathrm{Wet}\;\mathrm{weight}-\mathrm{Dry}\;\mathrm{weight}}{\mathrm{Dry}\;\mathrm{weight}}\times 100\%$$

The topological assessments of the PCM and CM materials were performed with atomic force microscopy (AFM) by using a scanning probe microscope (NT-MDT, NTEGRA Prima, NT-MDT Spectrum Instruments, Moscow, Russia) with a golden silicon probe (tip height: 14–16 µm) under contact mode at the sample dimension 2 µm × 2 µm. Resistivity of the PCM material was determined via electrochemical impedance spectroscopy (IS) (Solatron 1260, Ametek Scientific Instruments, Berwyn, PN, USA) with a frequency range between 1 and 100 kHz and AC amplitude of 2900 mV. The samples were coated with silver paint prior to testing and analysed using ZView 2 software (AMETEK Scientific Instruments, USA, Version 2.5b). The resistivity value of the PCM materials was calculated using the following Equation (2):(2)$$\mathrm{Resistivity}\;\left(\rho \right)=\frac{\mathrm{RA}}{L}$$
where R (Ω) is the resistance value obtained from IS, A is the area (cm^2^), and L is the thickness (cm) of the PCM materials. Additionally, Jonscher power law was also obtained using the following Equation (3): (3)$$\sigma_{ac}=\sigma_{dc}+A\omega^{s}$$
where σ_ac_ is the AC conductivity calculated from the ρ in Equation (2), σ_dc_ is the DC conductivity, ω is angular frequency, A is the pre-exponential factor, and s is the frequency exponent.

## 3. Results and Discussion

### 3.1. Characterizations of PF Resin

The yield of LEFB was 61.8%, and it contained 25.0% holocellulose and 10.4% lignin (Table 1). EFB was degraded into small fragments, after undergone liquefaction. Figure 1a shows that the FTIR spectrum of synthesized PF and commercial PF were almost identical. The broad peak at 3300 cm^−1^ corresponded to a hydroxyl group (OH-) derived from phenol (commercial PF) and lignin (synthesized PF). The significant difference in the spectrum is the presence of peaks at 2800 and 2900 cm^−1^ for synthesized PF that can be attributed to methyl (CH_2_) and methylene (CH_3_) stretching. The peaks occurred mainly due to the presence of lignin and cellulose in the EFB fibre. The pronounced peak at around 1600 cm^−1^ corresponds to the alkenes from the aromatic ring of phenol [20], and, additionally, from lignin. The peaks at 1476 and 1482 cm^−1^, and 1450 cm^−1^ could be attributed to the methylene bridges at the para-para position and ortho-para position, respectively [21]. The formation of methylene bridges is a result of the condensation of methylol groups during the resinification process. The commercial PF resin showed a peak at around 1267 cm^−1^, which represents C–O–C linkage [22]; meanwhile, in synthesized PF, the peak at 1221 cm^−1^ belongs to asymmetric stretches of C–C–OH [23]. The aliphatic hydroxyl was represented by peaks at around 1020 cm^−1^ [24]. It was found that the C–C–OH contributed by the uncondensed methylol groups was initiated from the addition reaction of phenol and formaldehyde, which are stable in alkaline conditions. Usually, peaks of around 1000–650 cm^−1^ are due to plane assignments of C–H vibrations [25] that happen at approximately 960 cm^−1^ in commercial PF, and 820 and 750 cm^−1^ in synthesized PF.

Figure 1b is a DSC thermogram with two peaks of synthesized (62 and 107 °C) and commercial PF resin (96 and 120 °C). The first was attributed to the formation of hydroxymethyl phenol from the addition reaction of formaldehyde at the aromatic ring, and the second peak was attributed to the crosslinking during curing at T_cure_ [26] or the glass transition temperature (T_g_) [27]. The synthesized PF resin (12.4 J g^−1^) gave a higher-temperature crosslinking reaction than the commercial resin (5.1 J g^−1^). The synthesized resin had a higher degree of crosslinking and more methylene bridges [28]. This crosslinking would also be affected by the T_cure_ or T_g_ value. As the T_cure_ or T_g_ is low, the degree of crosslinking is high.

The stages of TGA curves of commercial and synthesized resins are depicted in Figure 1c,d. The weight loss of commercial and synthesized PF resins was 63.0 and 36.1%, respectively. The stage around 30–120 °C was due to the loss of water, while the degradation stage at 121–140 °C was due to the loss of formaldehyde and methanol [29,30]. Between 150 and 380 °C, the loss of weight was mainly due to the release of carbon monoxide and carbon dioxide. Furthermore, the synthesized PF also released C_1_ components at this temperature range. Compounds that are released at temperatures below 380 °C are from polyoxymethylenes, hemiacetals, and methoxyhemiacetals, which occur due to the cleavage of the methylene bridge at the ortho-para and para-para positions [15]. The peak at around 450–550 °C was due to the pyrolytic decomposition of aromatic compounds that are derived from phenol and/or lignin.

### 3.2. Phenolic Uptake of PCM Materials

Our results show that the PF concentration influenced the incorporation of the PF resin into the CM. The PF uptake was higher in PCMW (73–81%) than PCMD (9–17%) for all the resin concentrations (Figure 2). As the PF concentrations increased, the phenolic uptake decreased. The highest PF uptake was found at 5% PF at the values of 81% and 17% for PCMW and PCMD, respectively. At higher values than 15% PF, the PF uptake did not change significantly, thus revealing this to be the maximum concentration for impregnation.

The PF uptake in PCMW is almost two times higher than in PCMD. The huge difference in phenolic uptake was thus due to the conditions that caused different impregnation mechanisms (Figure 3). Under dry conditions, the mechanism involved is absorption. The cohesive forces between PF resin and CM led to high surface tension [31], causing pore closure of the PCMD [32] during penetration, thus restricting the absorption process and reducing phenolic uptake. Under wet conditions, the mechanism involved is solvent exchange [33] between water and resin. The different concentration gradient and formation of hydrogen bonds between water and CM led to the high PF resin uptake in PCMW.

### 3.3. CP-MAS NMR Characterization of PCM Materials

The molecular interactions of PF and PCM can be observed in the CP-MAS NMR spectrum (Figure 4). The CM shows the six common signals of the anhydroglucose unit at around 55–110 ppm, see also Hashim et al. (2020) [34]. These signals are still present in the PCM materials, with an additional three signals that derived from the PF resin. The peak at 150 ppm belonging to OH-substituted aromatic carbons, and the peaks around 120-130 ppm to non-substituted and C-substituted aromatic carbons. Another signal at around 35 ppm refers to methylene bridge carbons. These additional three peaks are the same as those reported in [35], which confirmed that the PF resin had undergone crosslinking during curing. Additionally, the intensity of the peak at around 130 ppm increased as the PF concentrations increased, as is clearly depicted in the PCMW materials.

### 3.4. Water Absorption of CM and PCM Materials

Table 2 shows the water absorption of the CM, PCMD and PCMW samples. The CM had the highest water absorption (175.1%) due to the hydrophilic properties of CM. The water absorption of the PCMD and PCMW materials decreased as the concentration of the PF resin increased. This was due to the formation of ether linkages during the curing process that consequently inhibited water from forming hydrogen bonds with CM [36,37]. The PCMW shows a lower water absorption because of the higher PF uptake. The formation of methylene bridges within the PF resin molecules caused it to be highly crosslinked and thus more rigid, which also restricted water absorption.

### 3.5. Thermal Properties of PCM Materials

The TGA curves (Figure 5a) show that CM had two degradation stages. The first degradation was attributed to the release of water (84.24%), which occurred at 30–120 °C. The second degradation (230–365 °C) was attributed to the degradation of the cellulose component, and there was only 6.84% weight loss at this stage. The heating process caused the cleavage of the glycosidic bond, thus decomposing cellulose into smaller fragments, i.e., carbon dioxide, water, and other hydrocarbon derivatives [38]. The DSC curve in Figure 5b shows a broad exothermic peak that is correlated with the TGA peak at 105 °C. The high water removal is due to the abundance of hydroxyl groups in the cellulose membrane. The hydroxyl groups in the amorphous domain of cellulose have higher tendency to absorb water [39].

The TGA curves of PCMD and PCMW are depicted in Figure 6. For the PCM samples, only two stages of degradation were present after the water loss. The first degradation occurred at 230–365 °C and the second degradation occurred at 365–425 °C. The weight loss between PCMD and PCMW was significantly different, due to the difference in PF uptake between the samples. As a result, PCMD gives 47–49% and PCMW gives 28–40% weight loss during heating up to 365 °C. The weight loss at the second degradation stage is due to the decomposition of the methylene-bridged aromatic PF resin [15]. At this stage, PCMD shows no weight loss due to low PF resin uptake. For PCMW, the weight loss was 6–9%, and the value increased with the increasing PF resin concentration. During the curing process of PCM samples, there were two interactions: (i) within the PF resin and (ii) between the PF resin and CM [40]. The interaction within the PF resin consisted of the condensation reaction of phenol and the methylol groups [41], while the interaction between the PF resin and CM also caused the formation of ether linkages with the polysaccharide [36,37]. PCMW had high PF uptake and, thus, allowed more interaction between the PF resin and CM. The weight loss values were directly proportional to the PF resin concentrations.

Two exothermic peaks were seen in the DSC curves for both PCMD and PCMW, as shown in Figure 7. The first exothermic peak at 30–110 °C was due to the release of water. The second peak appeared at 240–248 °C, which corresponded to the T_g_ value of the PCM. The results support that the PF had good compatibility with the CM—there was a single T_g_ in every sample [42], usually at around 240–248 °C, except for PCMW10. The T_g_ for PCMW10 was found to be at 220 °C, likely due to the formation of defects, voids, or aggregation, during the curing process [41]. The T_g_ values of PCMD and PCMW decreased as the concentration of PF resin increased. The PF resin hindered homogeneous heat flow during the DSC analysis, hence lowering the T_g_ value [42].

### 3.6. Surface Roughness of CM and PCM Materials

A topological study was carried out to examine the surface roughness of the samples. The images show differences in the colour gradient: the dark- and bright-coloured gradients symbolize the short peak (valley) and high peak (high surface), respectively. Figure 8 shows that the CM had bumpy surfaces with irregular peak heights. The root mean square roughness (RMS) of CM was 10.07 nm.

In addition, S_q_ value was obtained from the AFM topography as shown in Figure 9. The S_q_ of the PCMD materials increased as the concentration of PF resin increased (5–20 wt.%), presenting the values 20 (a), 60 (c), 66 (e), and 75 (g) nm. Similar patterns can be observed for PCMW. The S_q_ of PCMW materials were 35 (b), 63 (d), 71 (f), and 37 (h) nm at 5, 10, 15, and 20 wt.%, respectively. The S_q_ of PCMW was higher because the thick and swollen CM enhanced the impregnation of the PF resin. The PF resin in CM was aggregated, shrunken, hardened, and formed an irregular surface during curing. As a result, different peak heights were produced [43]. However, the S_q_ of PCMW dropped at 20 wt.%. This is probably because of the heterogeneity in crosslinked samples, since the surface roughness should increase as the sample is more crosslinked.

### 3.7. Electrical Analysis of the PCM Materials

Impedance spectroscopy (EIS) showed that the sample conditions significantly influenced the log-log resistivity dependence on resin concentrations (Figure 10a). CM had the lowest resistivity (10^7^ Ω), followed by PCMW (10^8^ Ω) and PCMD (10^9^ Ω). This means that dry impregnation of the PF resin gave higher resistivity than the wet mechanism. The resistivity of PCMW and PCMD materials increased proportionally with the increase in PF resin concentrations, up to 15 wt.%; PCMD15 and PCMW15 materials show the highest resistivity, with values of 1.71 × 10^9^ Ω and 1.76 × 10^8^ Ω, respectively. As the PF concentration was increased to 20 wt.%, the resistivity of PCMD and PCMW decreased. This was due to its irregular surface roughness with high peaks (bumpy surface). Consequently, the electrons tend to move by tunnelling [44].

Figure 10b shows the log-log variation in σ_ac_ as a function of frequency. The obtained σ_ac_ of CM and PCM materials decreased upon decreasing the frequency, and increased upon increasing the frequency. PCMD5 gave the lowest σ_ac_ values among all the samples. As the σ_ac_ becomes independent, the curve can be split into two parts. The first region occurred at the plateau curve, and the second region is the increase at higher frequency. This phenomenon obeys the characteristics of Jonscher’s power law, with σ_ac_ being directly proportional to ω^s^. 

## 4. Conclusions

In this study, the fabrication of PCM materials was successfully conducted by PF impregnation of CM. This resulted in a significant influence on the physical, chemical, thermal, and electrical properties of PCM materials. The phenolic uptake in PCM materials was dependent on the CM conditions and PF concentrations. The amount of PF in CM affected the crosslinking of the samples, and, therefore, also their thermal stability. As the CM was treated with PF, the hydrophilic properties of the PCM materials were reduced. Subsequently, as the PF concentrations increased, the surface roughness and resistivity of the PCM materials also increased. All PCM materials obey Jonscher’s power law that describes the electrical conductivity of disordered materials. 

## Figures and Tables

**Figure 1 membranes-12-00106-f001:**
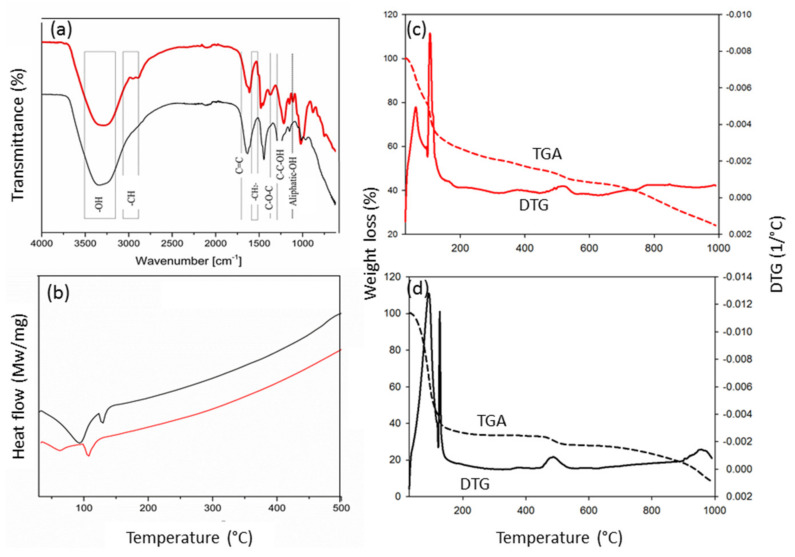
Analysis of commercial (−) and synthesized (−) phenolic resin (PF). (**a**) Fourier transform infrared (FTIR) spectrum shows that functional groups present in commercial PF were also found in synthesized PF, (**b**) the T_g_ of synthesized PF (106 °C) was lower than commercial PF (119 °C), (**c**) thermogravimetric analysis (TGA) and derivatives thermogravimetry (DTG) curves of synthesized PF, and (**d**) TGA and DTG curves of commercial PF.

**Figure 2 membranes-12-00106-f002:**
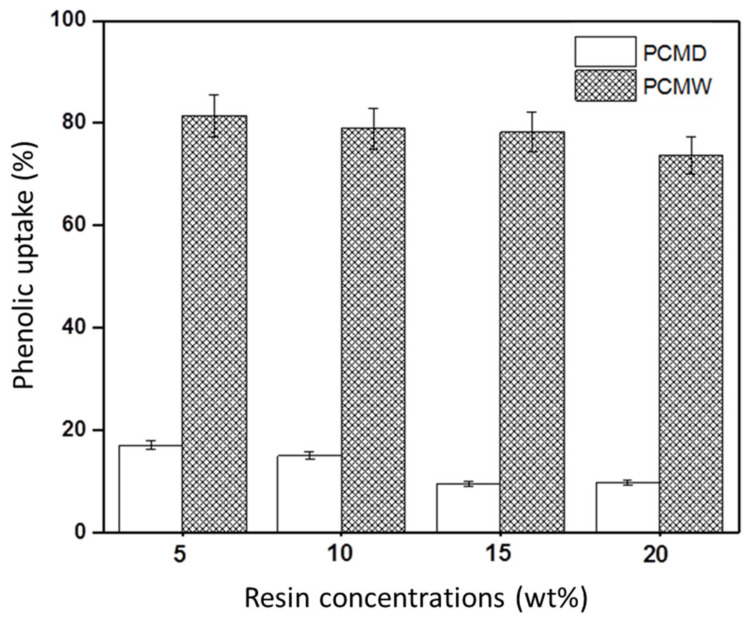
Synthesized PF uptakes at different concentrations in wet phenolic cellulose membrane (PCMW) and dry phenolic cellulose membrane (PCMD).

**Figure 3 membranes-12-00106-f003:**
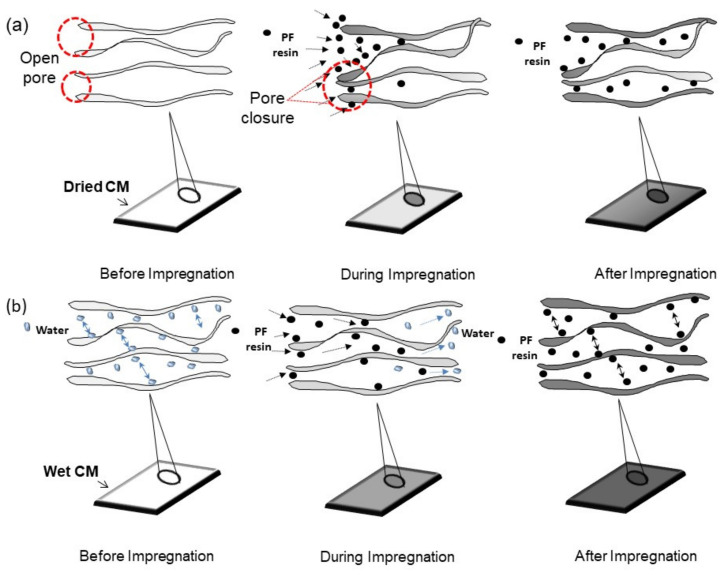
Schematic diagrams for PF uptake in (**a**) dried and (**b**) wet membranes.

**Figure 4 membranes-12-00106-f004:**
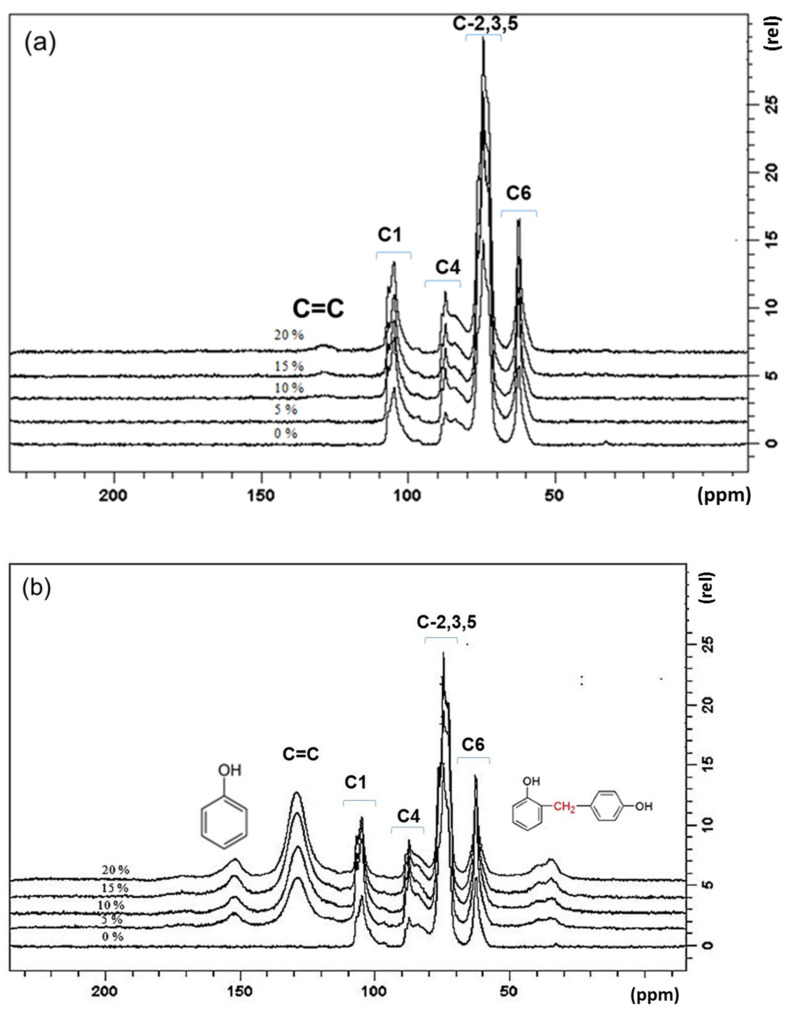
Cross polarization magic-angle-spinning nuclear magnetic resonance (CP-MAS NMR) of (**a**) PCMD and (**b**) PCMW materials.

**Figure 5 membranes-12-00106-f005:**
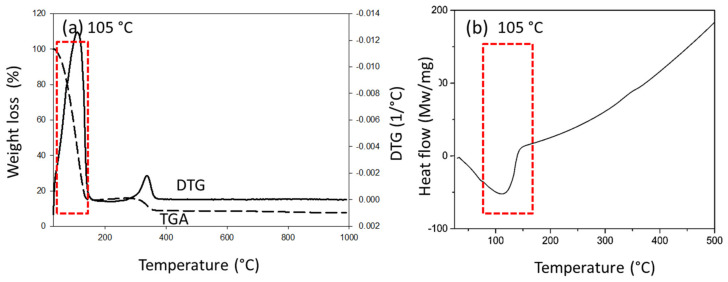
(**a**) Two degradation stages and (**b**) a single exothermic peak at 105 °C.

**Figure 6 membranes-12-00106-f006:**
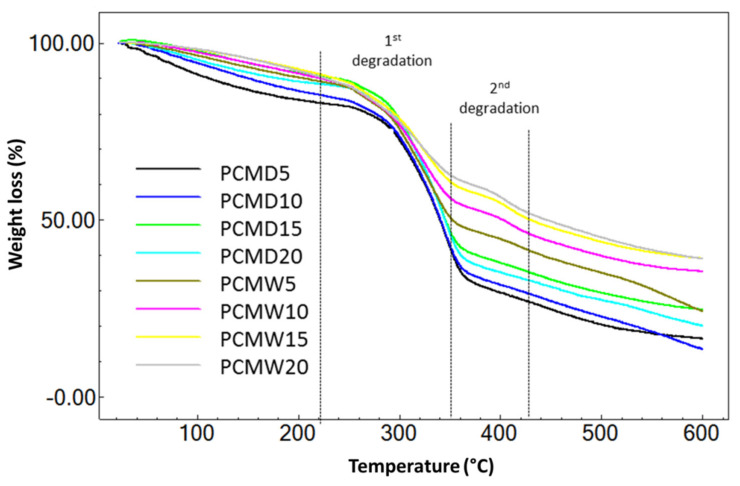
Weight loss of PCM, influenced by the impregnation conditions.

**Figure 7 membranes-12-00106-f007:**
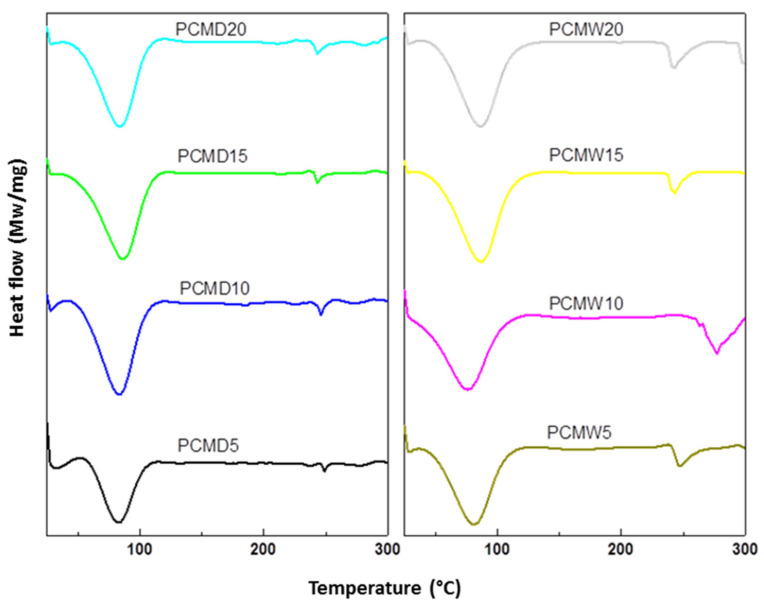
Normalized differential scanning calorimetry (DSC) curves for PCM materials.

**Figure 8 membranes-12-00106-f008:**
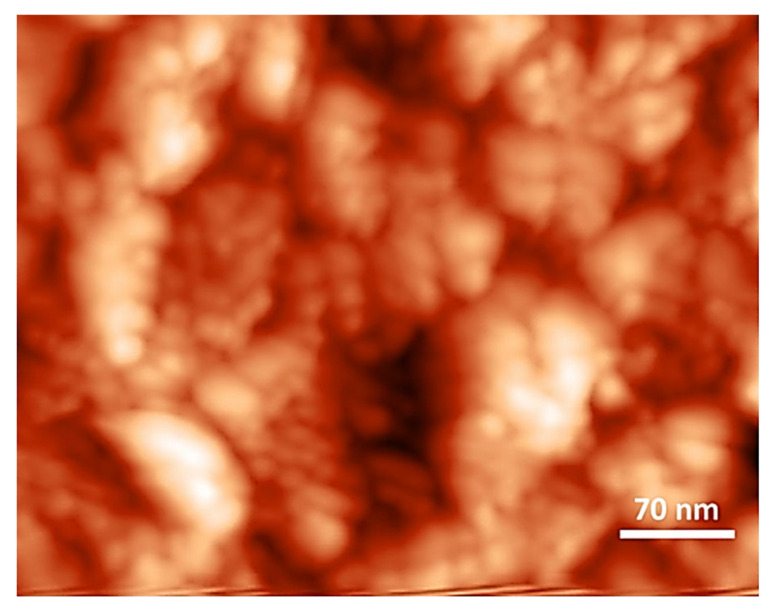
AFM topography at 2 µm × 2 µm of cellulose membrane (CM).

**Figure 9 membranes-12-00106-f009:**
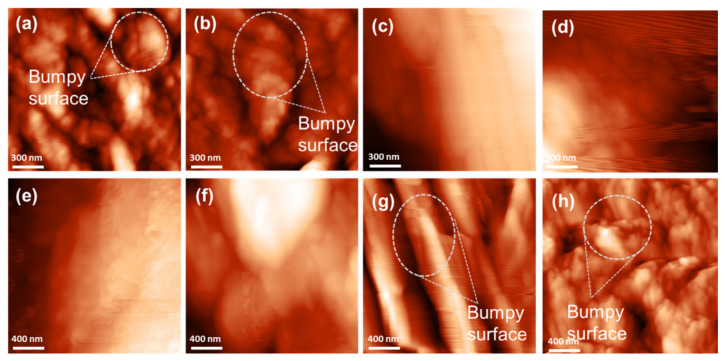
AFM topography at 2 µm × 2 µm (**a**) PCMD5, (**b**) PCMW5, (**c**) PCMD10, (**d**) PCMW10, (**e**) PCMD15, (**f**) PCMW15, (**g**) PCMD20, and (**h**) PCMW20.

**Figure 10 membranes-12-00106-f010:**
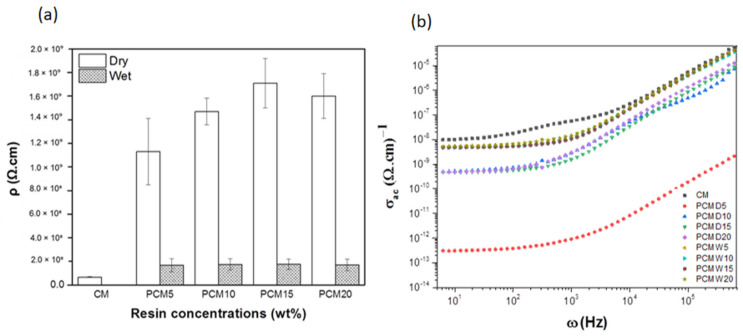
(**a**) Resistivity of PCMW and PCMD materials at different resin concentrations and (**b**) frequency-dependent AC conductivity of CM and PCM materials.

**Table 1 membranes-12-00106-t001:** Compositions of oil palm empty fruit bunch (EFB) fibre and after liquefaction.

Compositions	EFB Fibre (%)	LEFB (%)
Holocellulose	70.4	25.0
Lignin	25.6	10.4
Yield	-	61.8
Residue	-	38.2

**Table 2 membranes-12-00106-t002:** Water absorption of phenolic cellulose membrane (PCM) materials.

PF Concentrations (wt.%)	Water Absorption (%)	
	PCMD	PCMW
Cellulose membrane (CM)	175.1 ± 1.7	
5	61.8 ± 4.0	56.8 ± 0.6
10	43.8 ± 5.1	23.4 ± 3.0
15	45.9 ± 0.7	16.3 ± 1.4
20	43.1 ± 0.2	13.0 ± 0.4

## Data Availability

The data presented in this study is presented directly in this article.

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
