# Peer review of "Effect of Impregnated Phenolic Resins on the Cellulose Membrane for Polymeric Insulator"

_membranes, 2022, doi:10.3390/membranes12020106_

Round 1

Reviewer 1 Report

Dear Author, 

The manuscript provide some interesting information about the properties of electrically insulating cellulose membranes. However, there are some weaknesses in the section of electrical properties. In the DETA plot the y-axis should be expressed in Log, since the decrease of the resistivity as function of the frequency is not fully observed. Also, the decrease of the resistivity with the increase of the frequency is not neccessary attributed "to effect of water/moisture in CM and PCM composites". This behaviour can be related with the insulating nature of the material, Please verify the Jonsher equation and the dependece of the conductivity (1/R) as function of the frequency. So, the discussion should be improved. 

Author Response

Comment 1: The manuscript provides some interesting information about the properties of electrically insulating cellulose membranes. However, there are some weaknesses in the section of electrical properties. In the DETA plot the y-axis should be expressed in Log, since the decrease of the resistivity as function of the frequency is not fully observed. Also, the decrease of the resistivity with the increase of the frequency is not necessarily attributed "to effect of water/moisture in CM and PCM composites". This behavior can be related with the insulating nature of the material, please verify the Jonscher equation and the dependence of the conductivity (1/R) as function of the frequency. So, the discussion should be improved. 

Response 1: Refer subtopic 3.7 Electrical Analysis of PCM Composites

We thank the reviewer for pointing out the error. We apologize for the wrong X and Y-axis labels on the previous graphs. The presented graphs are actually the Log Resistivity Vs Frequency graphs and not Resistivity Vs Log Frequency graphs.

As suggested by the reviewer, we have prepared the Jonscher plot for the CM and PCM composites. Jonscher power law is used to describe the electrical conductivity of disordered CM and PCMs. For the Jonscher plot, we presented in log-log σac versus ω. Since this new addition looks similar to our previous resistivity dependent frequency graphs, we have removed the older graph to avoid redundancy.

These changes can be found in Figure 10 of the revised manuscript.

We hope this has made the presentation of the results clearer.

Reviewer 2 Report

The manuscript describes the investigation of impregnating regenerated cellulose with phenol-formaldehyde resin to form composites as insulators. It characterizes their chemical, physical, thermal, morphological, and electrical properties. However, no porous structure and transport properties relevant to membranes have been characterized. This reviewer believes that this manuscript does not fall in the scope of this journal. Some specific points as follows:

Figure 1, (c) TGA diffractogram   should be TGA derivative, please reverse the derivative curves with peak up

How much empty fruit bunch was liquified? Figure 1 FTIR spectra show the synthesized PF is the same as the commercial one, which indicates no substantial LEFB constituents presented in the synthesized PF resin. What is the lignin, cellulose percentages in the EFB? How those derivatives from cellulose do not show some bands on the FTIR spectrum of the synthesized PF resins? They do not have a ring of benzene.

Author Response

Comment 1: Some specific points as follows. Figure 1, (c) TGA diffractogram   should be TGA derivative, please reverse the derivative curves with peak up

Response 1: TGA derivatives which are also noted as DTG in Figure 1 (c) and (d) have been modified by reversing the curve upward. Additionally, the term of diffractogram has been amended to curves.

Comment 2: How much empty fruit bunch was liquified?

Response 2: Refer 3.1 Characterizations of PF resin.

EFB had been successfully liquefied with 61.8% yield. This information has been added under this subtopic in the first paragraph and is annotated as a comment in the revised manuscript.

Comment 3: Figure 1 FTIR spectra show the synthesized PF is the same as the commercial one, which indicates no substantial LEFB constituents presented in the synthesized PF resin.

Response3: In subtopic 3.1 Characterizations of PF resin, we have included the statement below:

The significant difference in the spectrum is the presence of peaks at 2800 and 2900 cm−1 for synthesized PF, which attributed to methyl (CH2) and methylene (CH3) stretching. The peaks are owned mainly by lignin and cellulose in the EFB fiber.”

This states that the spectrum of synthesized PF is slightly different with commercial PF, with additional peaks of CH2 and CH3 that derived from lignin.

Comment 4: What is the lignin, cellulose percentages in the EFB?

Response 4: Refer 3.1 Characterizations of PF resin.

EFB contains 25.6% of lignin and 70.4% of holocellulose. Meanwhile, for the LEFB, it consists of 10.4% of lignin and 25.0% of holocellulose. We had carried out the analysis for this question, but the results were not presented earlier. However, due to the concern of the reviewer, we have added the information in Table 1.

Comment 5: How those derivatives from cellulose do not show some bands on the FTIR spectrum of the synthesized PF resins? They do not have ring of benzene.

Response 5: Refer subtopic 3.1 Characterizations of PF resin.

After undergoing liquefaction, the cellulose fragments that could be found was C-O and CH. These fragments could also found in lignin. FTIR analysis unable to distinguish the functionalities belongs to which constituents if they have the same functionalities. Therefore, will lead to overlapping peaks.

However, in our writing, we have discussed on those peaks that presence in synthesized PF in general.

Commercial PF resin shows a peak around 1267 cm−1 which represents C-O-C linkage [22] meanwhile in synthesized PF the peak at 1221 cm−1 belongs to asymmetric stretch of C-C-OH [23]à The C-OH here could be attributed by cellulose and also lignin.

Usually peaks around 1000-650 cm−1 are assignment of out of planes of C-H vibrations [25] that happen approximately at 960 cm−1 in commercial PF and 820 and 750 cm−1 in synthesized PF à The peaks at 820 and 750 cm−1 could be contributed by cellulose and also lignin.

The discussion in this part mainly focuses on the evidence that synthesized PF resin has been successfully produced and having same functionalities as commercial PF.

Reviewer 3 Report

The paper is sound. Well written and discussed. Some small shortcomings to enhance the quality: 

  • The novelty of the present investigation should be stressed.
  • what is the porosity of the cellulose membrane? particle size? As resin uptake depends on it, these characteristics need to be provided.
  • Figure 1 cover text should contain info about curves abbreviation, it is not clear which curve is TGA and DTG
  • Conclusions should be more specific on obtained results
  • Abstract also needs to be enhanced with specific information.

Author Response

Comment 1: The novelty of the present investigation should be stressed.

Response 1: We have rewritten the last paragraph in the Introduction, to address the knowledge gap and highlight the novelty of this work.

Previous works have used cellulose as insulator materials due to its thermal stability and easy processing. However, because of its hydrophilic properties, the performance could drop and become failure. Therefore, we proposed to treat the cellulose insulators (here we presented in membrane) with phenolic (PF) resin in order to overcome the problem. We tried to understand the impregnation mechanism of PF into the cellulose membrane (CM). We believe, this mechanism and the PF uptakes will affect the phenolic cellulose membrane (PCM) insulators. We have added this commentary to the revision and is annotated as a comment in the revised manuscript.

Comment 2: What is the porosity of the cellulose membrane? particle size? As resin uptake depends on it, these characteristics need to be provided.

Response 2:  The porosity of the cellulose membrane is not included in the manuscript because it was published in our previous publication In-depth Characterization of Cellulosic Pulps from Oil Palm Empty Fruit Bunches and Kenaf Core, Dissolution and Preparation of Cellulose Membranes. SNAS Hashim, BAZ Norizan, KW Baharin, S Zakaria, CH Chia, A Potthast, S Schiehser, M bacher, T Rosenau, SNS Jaafar 2020. Cellulose Chemistry and Technology 54 (7-8), 643-652. 

 From the study, it was determined that the pore size of the membrane was between 1-2000 µm, in addition to being dense and well distributed.  

Here in this manuscript, we tried to look on different factor of resin uptake. We believe, the membrane conditions (wet and dry) and PF resin concentrations, could influenced to the resin uptake by using the fabricated membrane (with pores size of 1-2000 µm).

We have clarified this fact under subtopic 3.2 Phenolic Uptake of PCM Composites.

Comment 3:  Figure 1 cover text should contain info about curves abbreviation, it is not clear which curve is TGA and DTG

Response 3: We thank the reviewer for this comment as it was a confusion that was also commented on by another reviewer. The figure and caption have been amended by specifying TGA and DTG curves which has improved its clarity.

Comment 4:  Conclusions should be more specific on obtained results

Response 4: The conclusions section has been modified to focus on the results obtained in the reported study.

Comment 5:  Abstract also needs to be enhanced with specific information.

Response 5: The abstract writing has been improved and recommended by the reviewer.

Round 2

Reviewer 2 Report

None

Reviewer 3 Report

The Paper was corrected. It can be accepted in the present form.